# Soil Enzyme Activity Behavior after Urea Nitrogen Application

**DOI:** 10.3390/plants11172247

**Published:** 2022-08-29

**Authors:** Benjamin Davies, Jeffrey A. Coulter, Paulo H. Pagliari

**Affiliations:** 1Famine Early Warning Systems Network, Washington, DC 20003, USA; 2Department of Agronomy and Plant Genetics, University of Minnesota, Saint Paul, MN 55108, USA; 3Southwest Research and Outreach Center, University of Minnesota, Lamberton, MN 56152, USA

**Keywords:** FDA, fluorescein diacetate, PP, pre-plant application, PM, maize physiological maturity, RN, (RN) nitrogen recommended rate, R1, silking stage of maize phenological development, SB-M-M-M, soybean-maize-maize-maize, Sp, two-way split application, TSp, three-way split N application, V6, six-leaf collar stage of maize phenological development

## Abstract

Understanding how fertilizer application (particularly N, the most used chemical fertilizer worldwide) interacts with soil microbes is important for the development of best management practices that target improved microbial activity to enhance sustainable food production. This study was conducted to determine whether urea N rate and time of application to maize (*Zea mays*) influenced soil enzyme activity. Enzyme activity was determined by monitoring fluorescein diacetate (FDA) hydrolysis, ß-glucosidase, acid-phosphomonoesterase, and arylsulfatase activities. Experiments were conducted from 2014 through 2016 to compare single (fall or spring applications) and split applications of N at varying N rates under irrigation (Becker) and rainfed conditions (Lamberton and Waseca) in MN, USA. Nitrogen rates varied by location and were based on University of Minnesota guidelines. Soil samples were collected seven times each season. Nitrogen application split into two applications increased FDA activity by 10% compared with fall and spring applied N at Waseca. Fall or spring N application decreased arylsulfatase activity by 19% at Becker and by between 13% and 16% at Lamberton. ß-Glucosidase and acid-phosphomonoesterase activities were unaffected by N application. Sampling time and year had the greatest impact on enzyme activity, but the results varied by location. A negative linear relationship occurred between FDA and ß-glucosidase activity at all three sites. In summary, urea N application had small effects on enzyme activity at the sites studied, suggesting that some form of organic N could be more important than the ammonium provided by urea.

## 1. Introduction

Current estimates indicate that the global population could reach 10 billion people by 2050. This increase in population will require a much more efficient and sustainable food production system. Our food production system relies heavily on chemical inputs, such as N-containing fertilizer and pesticides, for maximum plant yield. In a more integrated food production system, the soil microbial communities will likely play a significant role in achieving and maintaining system sustainability. Soil organisms can play a pivotal role in nutrient cycling. Soil microflora (bacteria and fungi) and fauna (protozoa and invertebrates such as nematodes, mites, and earthworms) can influence the availability of nutrients for crop production through the decomposition of crop residues, mineralization and immobilization of nutrients, biological N fixation, and bioturbation. Mineralization or immobilization of nutrients by soil microbes influences temporal nutrient availability, soil nutrient status, and net productivity of agroecosystems [1]. Soil microbial biomass is regarded as an early indicator of changes in soil fertility and agroecosystem properties [2]. The impact of N fertilization on soil microbial biomass and activity is still not fully understood. Understanding how fertilizer application (particularly N, the most used chemical fertilizer worldwide) interacts with soil microbes is of utmost importance for the development of best management practices that target improved microbial activity for efficient, sustainable food production.

The adverse effects of long-term N fertilizer application on soil enzyme activity was reported from a 20-year fertilization regime in a vegetable greenhouse production system [3] because the application of 300 and 600 kg N ha^−1^ decreased α- and β-glucosidase activities compared with 75 Mg ha^−1^ of composted manure [3]. However, a meta-analysis of long-term trials from around the world concluded that the application of mineral fertilizers (N, P, and K fertilizers) increased microbial biomass by 15% compared to no fertilization [4]. The addition of mineral fertilizer also increased soil organic C by 12% compared with no fertilization and was a major factor contributing to increased microbial biomass [4]. The duration of experiments has also been reported to affect the response of soil microbial biomass, with the greatest increase in microbial activity in experiments longer than 20 years [4]. In contrast, a different meta-analysis reported that microbial biomass declined 15% with N fertilization (the decline in microbial biomass increased as N load increased) [5].

The activity of soil enzymes involved in the degradation of soil organic matter can be used to provide an early indication of changes in soil health as a result of changes in soil management [6,7,8]. The hydrolysis of FDA has been found to represent a wide range of soil enzyme activity, as many enzymes can hydrolyze FDA, including lipases, esterases, and proteases [9]. To our knowledge there is no research that closely looks at the effects of N fertilizer addition on FDA activity. β-Glucosidase activity is a useful enzyme for soil quality monitoring because of its central role in the enzymatic degradation of cellulose, the main component of plant polysaccharides, and soil organic matter cycling [7]. ß-Glucosidase activity can provide a reflection of past biological activity and the capacity of soils to stabilize organic matter [10]. To our knowledge, there have been few studies that compared the effects of N fertilizer rate and timing of application on β-glucosidase in maize cultivated under continuous production. A greenhouse vegetable trial reported that β-glucosidase activity increased when 75 Mg ha^−1^ of composted manure was applied at planting followed by a sidedress of 138 or 276 kg N ha^−1^, due to a greater turnover rate of soil C [3]. This supports the findings of the meta-analysis by Geisseler and Scow [4] and Jian et al. [11], where the application of N fertilizer increased β-glucosidase activity by 11 to 15%, likely also as a result of improved soil C. Since microbial activity is influenced by changes in soil organic C, increased crop productivity should increase the amount of crop residue returned to the soil, thus increasing soil organic matter content over time [4]. In contrast, an incubation study using soils from 28 ecosystems across North America where no C inputs were added concluded that β-glucosidase activity decreased by 12% and soil microbial biomass decreased by 35% when N was added [12]. The addition of N may lead to increased soil C sequestration rates, due to a shift in the metabolic capabilities of soil bacterial communities and the favoring of microbial communities less capable of decomposing more recalcitrant soil C pools [12].

Similar to β-glucosidase activity, soil phosphomonoesterase increases with the addition of organic matter, likely as a result of increased organic P, which must be first mineralized before it can be used by microbes and plants [13,14]. In a rice (*Oryza sativa*) wheat (*Triticum aestivum*) rotation system on a sandy loam soil in China, the application of different N rates and sources increased phosphomonoesterase activity by 8 to 71% compared with no fertilization [15]. In addition, by reducing traditional N fertilizer doses by 20% and replacing 50% of N fertilizer with organic matter, phosphomonoesterase activity increased by 35 to 74% compared with traditional N fertilizer doses [15]. Nitrogen fertilization was reported to have had no effect on soil phosphomonoesterase activity in a grazed-pasture system [16]. Similarly, a field experiment assessing different rates of urea-N and broiler litter-N on phosphomonoesterase activity reported that alkaline phosphomonoesterase activity increased with increasing rates of broiler litter but not with increasing rates of urea [17]. In contrast, incubation studies have reported an increase in soil phosphomonoesterase activity with N fertilization [18]. The application of increasing N fertilizer (0 to 300 mg N kg^−1^ applied as NO_3_^−^-N or NH_4_^+^-N) increased phosphomonoesterase activity but only if the N source was nitrate, whereas ammonium had no effect [18]. The lack of increased phosphomonoesterase activity with ammonium application was attributed to microbial reallocation of C to biomass or enzyme production [19]. The lack of a response with ammonium suggests that either microbes are mineralizing less C from protein sources, due to reduced protease production, or that ammonium restricts microbial respiration rates [20]. Allison and Vitousek [20] also found that β-glucosidase and acid phosphomonoesterase activities increased in treatments where only carbon and nitrogen were added, while the activities of enzymes such as acid phosphomonoesterase and glycine aminopeptidase declined in response to ammonium and phosphomonoesterase additions.

Arylsulfatase is secreted to release soil available S from organic S (mainly ester sulfates); this enzyme can be a useful indicator of soil health [21]. Similar to the activities of β-glucosidase and phosphomonoesterase, arylsulfatase is highly correlated with soil organic C content [22,23]. A three-year greenhouse study compared the application of different rates of ammonium sulfate and organic fertilizer and reported that arylsulfatase activity was greatest when the soil organic matter level was high and also when 540 kg N ha^−1^ yr^−1^ was applied as compost [24]. Only a limited number of studies have investigated the effect of N fertilizer on arylsulfatase activity. Soil acidification due to N addition decreased arylsulfatase activity by 40%, while the addition of water increased arylsulfatase activity in a semi-arid grassland [21]. In contrast, a field study in Iowa reported that N fertilization had no effect on arylsulfatase activity [23]. The effect of N and water addition on arylsulfatase activity highlights the complex relationship of soil enzyme activity with the hydrolysis of urea.

Over the years, correlations have been established on the effect of tillage, pH, and residue management on soil enzyme activities involved in C, N, P, and S cycling in soils [22,25]. More information is needed to better understand the complex relationship between soil enzyme activity and N fertilizer management. In particular, little research has explored the effect of different N rates and timing of application in maize on soil enzyme activities. This study was designed to provide an opportunity to assess how the application of different urea-N rates and timing of application impacted soil enzyme activities under a soybean [*Glycine max* L. (Merr.)]-maize-maize-maize (SB-M-M-M) production system. Our hypothesis is that the addition of N fertilizer would stimulate microbial growth, resulting in an increase in the activity of the enzymes targeted in this research. The enzyme activities selected for study are responsible for C turnover (β-glucosidase), organic P turnover (acid phosphomonoesterase), and organic S turnover (arylsulfatase); in addition, fluorescein diacetate (FDA) was studied as a surrogate for overall microbial activity.

## 2. Materials and Methods

### 2.1. Site Description and Experimental Design

Field experiments were conducted at University of Minnesota Research and Outreach Centers (ROCs) near Lamberton, MN (44°24′ N, 95°30′ W), Waseca, MN (44°07′, 93° 52′ W), and Becker, MN (45°39′ N, 93°89′ W) from 2014 to 2016. All sites were planted to soybean in 2013 and to maize from 2014 to 2016. Soils were an irrigated Hubbard–Mosford loamy sand complex (Sandy, mixed, frigid Typic Hapludolls or frigid Entic Hapludolls) at Becker, with an organic matter (OM) level of 13 g kg^−1^; rainfed Normania loam (Fine-loamy, mixed, superactive, mesic Aquic Hapludolls) at Lamberton, with an OM level of 23 g kg^−1^; and rainfed Nicollet clay loam (Fine-loamy, mixed, superactive, calcareous, mesic Typic Endoaquolls) at Waseca, with an OM level of 36 g kg^−1^. The experimental design was a randomized complete block with four replications of 10 N fertilizer management treatments. Treatments were applied to the same plots in all years, making this a repeated measures design, and were analyzed according to the methodology of Pagliari et al. [26]. Plots were 4.6 m (six 76 cm rows) wide by 15 m long at Becker and Waseca and 6 m (eight 76 cm rows) wide by 12 m long at Lamberton, and the differences were due to the field equipment available at each ROC. In each year, the tillage system involved field cultivating in the spring prior to planting and stalk chopping followed by disk ripping in the fall after maize harvest. Phosphorus and potassium fertilizers were applied in the spring following soil test and University of Minnesota guidelines for maize production [27]. Pest management followed best management practices and varied by location and year.

Treatments in this study consisted of a non-N-fertilized control plus nine N fertilizer management treatments that varied in rate and time of N application as indicated in Table 1. The selected recommended N rate (RN) in 2014 for maize following soybean was 168 kg N ha^−1^ at Becker and 135 kg N ha^−1^ at Lamberton and Waseca [27]. In 2015 and 2016, the RN for maize following maize was 235 kg N ha^−1^ at Becker and 202 kg N ha^−1^ at Lamberton and Waseca. All rates were based on University of Minnesota guidelines [27]. The N source for fall applications was SuperU^®^ (46-0-0), a urea fertilizer with urease and nitrification inhibitors, and for pre-plant and in-season applications the N source was urea (46-0-0). Urea was selected as the N source for pre-plant and in-season N applications because it is a popular choice of fertilizer among growers in Minnesota [28]. SuperU^®^, which contains both urease and nitrification inhibitors, was selected as the N source for fall applications rather than urea to reduce the risk of N loss and to assess potential effects due to the urease and nitrification inhibitors.

All fertilizers were hand-applied and incorporated with tillage immediately after application in the fall and prior to planting. In-season applications were incorporated through irrigation at Becker. At Lamberton and Waseca, in-season applications of fertilizer were placed in 5 cm deep furrows in inter-rows manually made using a hoe which were closed with soil following fertilizer placement. The maize phenological stages selected for split applications of N were based on previous research on maize N requirement and uptake during the growing season [29]. The selected timings for split N application were aimed to synchronize N supply with maize N uptake. A complementary agronomic study on the impact of the timing and rate of these N fertilization treatments on maize grain yield and nitrogen use efficiency is reported elsewhere [30]. Davies et al. [30] also reports weather-related information for this trial.

### 2.2. Soil Sampling and Analysis

During each growing season, soil cores were collected from each plot from all treatments approximately one week before and one week after the V6 N application (Pre- and Post-V6, respectively); three weeks after the V6 N application (3wk-V6); one week before and one week after the R1 N application (Pre- and Post-R1, respectively); three weeks after the R1 N application (3wk-R1); and at maize physiological maturity (PM). The only exception was in 2016 at Waseca, when soil samples were not collected at Pre-R1, due to persistent wet soil conditions.

At each sampling event, six soil cores representing the plot area were collected from 0 to 15 cm depth, with three cores between maize rows two and three, and three cores between maize rows four and five. For each set of three cores, one core was in the row, one core was approximately 19 cm to the side of the row, and one core was approximately 38 cm to the side of the row. Samples were not collected within 1.5 m of the plot edges. Right after collection, the six soil cores for each plot were bulked and dried at room temperature (21 °C). After drying, samples were stored in paper bags until analysis.

### 2.3. Fluoroscein Diacetate Hydrolysis

Following the procedures described by Adam and Duncan [31], 1.0 g of air-dried soil (<2 mm) was mixed with 7.5 mL of potassium phosphate buffer, then 0.10 mL of FDA was added, and the sample was mixed and incubated at 37 °C for 3 h. After incubation, 5.0 mL of chloroform and methanol solution was added, capped, shaken, and centrifuged for 5 min at 3500× *g* rpm. The color intensity of the solution from each sample was measured by pipetting 250 µL of the filtrated samples into 96-well plates and measured using a spectrophotometer at a 490 nm wavelength (Epoch Biotek, Winooski, VT, USA). The data reported for FDA hydrolysis were adjusted to a per-hour hydrolysis to match the incubation time of the other enzymes.

### 2.4. β-Glucosidase, Acid Phosphomonoesterase, and Arylsulfatase Activities

The procedures for ß-1,4-glucosidase (hereinafter β-glucosidase), acid phosphomonoesterase (hereinafter phosphatase), and arylsulfatase were adapted from [32]. Briefly, 1.0 g of air-dried soil (<2 mm) was mixed with 4.0 mL of buffered solutions and 1 mL of substrate [p-nitrophenyl-*β*-D-glucopyranoside (for β-glucosidase), p-nitrophenyl phosphate (phosphatase), and p-nitrophenyl sulfate (arylsulfatase)], capped, and then incubated at 37 °C for 1 h. After incubation, buffer solutions (see [32] for solutions used for each enzyme) were added to stop the hydrolysis reaction, and the samples were centrifuged for 5 min at 3500× *g* rpm. After centrifugation, 250 µL of each centrifuged sample was transferred into a 96-well plate, and the color intensity was measured using a spectrophotometer (Epoch Biotek, Winooski, VT, USA) at wavelengths of 400 nm, 410 nm, and 420 nm for β-glucosidase, phosphatase, and arylsulfatase, respectively. Enzyme activity per kg of soil per hour was expressed as the release of p-nitrophenol per hour.

### 2.5. Statistical Analysis

Data were analyzed at *p ≤* 0.05 by repeated measures analysis of variance using the GLIMMIX procedure of SAS 9.4 [33] (SAS Institute, 2011). Treatment, sampling time, and year were considered fixed effects, and replication was considered a random effect. Sampling time and year were considered repeated measures, and the covariance structure that best fit the model for each parameter was assessed by checking the Akaike Information Criteria (AIC) among all possible covariance structures [23]. When appropriate, pairwise mean comparisons were made at *p ≤* 0.05 using the lines option in the GLIMMIX procedure of SAS. Data were analyzed separately by location due to many interactions among location and year, treatment, and sampling time.

A principal component analysis (PCA) was also carried out using data for soil enzyme activity, maize dry matter yield (DM), and soil nitrogen (NO_3_-N and NH_4_-N). Using the correlation matrix, principal components (factors) with eigenvalues >1 were retained. The correlation coefficients between the soil enzymes, maize dry matter yield, and soil nitrogen were interpreted as ordinary Pearson correlation coefficients.

## 3. Results

### 3.1. FDA

#### 3.1.1. Becker

Fluorescein diacetate hydrolysis was not significantly affected by treatment or interactions with treatment at Becker; however, the main effects of sampling time and year significantly affected FDA activity (Table 2). In 2014, FDA activity was greatest at 885 mg of fluorescein released kg^−1^ soil h^−1^, while FDA activity was similar in 2015 and 2016 at 695 and 651 mg of fluorescein released kg^−1^ soil h^−1^, respectively. Averaged across years and treatments, FDA was greatest at 3wk-V6 (931 mg of fluorescein released kg^−1^ soil h^−1^) and lowest at PM (640 mg of fluorescein released kg^−1^ soil h^−1^) (Table 3). Fluorescein hydrolytic activity prior to N application (mean = 703 mg of fluorescein released kg^−1^ soil h^−1^) was always significantly lower than post-N application (mean = 775 mg of fluorescein released kg^−1^ soil h^−1^) in the split treatments.

#### 3.1.2. Lamberton

Fluorescein diacetate hydrolysis was not affected by treatment at Lamberton; however, there were significant effects for sampling time, year, and the year × sampling time interaction (Table 2). In 2014, FDA hydrolysis Pre-V6 was 931 mg of fluorescein kg^−1^ soil h^−1^, significantly greater than that for other sampling times (Table 3). In 2014, FDA hydrolysis at Pre-V6 and Pre-R1 was significantly greater than that at Post-V6 and Post-R1 (Table 2), suggesting a potential negative effect due to urea application. In 2015, the effect of sampling time was less evident; however, FDA hydrolysis at Pre-V6 (656 mg of fluorescein kg^−1^ soil h^−1^) was significantly greater than that at PM (487 mg of fluorescein kg^−1^ soil h^−1^). In 2016, there were no differences in FDA activity between sampling times. Among all years, FDA was greater in 2014 (717 mg of fluorescein kg^−1^ soil h^−1^) than in 2015 and 2016 (mean = 559 mg of fluorescein kg^−1^ soil h^−1^).

#### 3.1.3. Waseca

Fluorescein diacetate hydrolysis was significantly influenced by treatment, sampling time, year, and the year × sampling time interaction (Table 2). The two-way split treatments (Sp75 and Sp100) had greater FDA hydrolysis (mean = 462 mg of fluorescein kg^−1^ soil h^−1^) than most of the other treatments receiving N fertilizer (mean = 417 mg of fluorescein kg^−1^ soil h^−1^) (Table 4). In 2014 and 2015, FDA hydrolysis was greatest at the Pre-V6 sampling time (618 mg of fluorescein kg^−1^ soil h^−1^), with less significant differences among samples collected during the growing season (Table 3). The only exception was a decrease in FDA activity from Post-R1 (482 mg of fluorescein kg^−1^ soil h^−1^) to PM (322 mg of fluorescein kg^−1^ soil h^−1^) compared with the other sampling times in 2015. In 2016, soil samples taken at Post-V6 and 3wk-V6 recorded 410 and 438 mg of fluorescein kg^−1^ soil h^−1^, greater than at later sampling times. Similar to the other locations, FDA hydrolysis at Waseca was greater at the beginning of each season and decreased over the growing season (Table 3). Fluorescein diacetate hydrolysis at Waseca decreased continuously in 2014, 2015, and 2016, with averages of 460, 435, and 383 mg of fluorescein kg^−1^ soil h^−1^, respectively.

### 3.2. β-Glucosidase Activity

#### 3.2.1. Becker

β-Glucosidase activity was not significantly influenced by treatment at Becker but was affected by sampling time, year, and the year × sampling time interaction (Table 2). In 2014, β-glucosidase activity across fertilization treatments was greater at the Pre-V6 sampling time (281 mg p-nitrophenyl released kg^−1^ soil h^−1^) and lower at the Post-V6 (average 158 mg p-nitrophenyl released kg^−1^ soil h^−1^) compared with the other samplings (mean = 203 mg p-nitrophenyl released kg^−1^ soil h^−1^) (Table 5). β-Glucosidase activity increased after the Pre-V6 sampling time; however, it was never as high as at the first sampling. In 2015, β-glucosidase activity fluctuated during the growing season; it was greatest at the Post-V6 samplings (306 mg p-nitrophenyl released kg^−1^ soil h^−1^) and lowest at the Pre-R1 sampling (198 mg p-nitrophenyl released kg^−1^ soil h^−1^). In 2016, β-glucosidase activity tended to be higher at the first two (mean = 193 mg p-nitrophenyl released kg^−1^ soil h^−1^) and last (221 mg p-nitrophenyl released kg^−1^ soil h^−1^) sampling times compared with the other sampling times (mean = 151 mg p-nitrophenyl released kg^−1^ soil h^−1^). Β-Glucosidase activity was the lowest at the three weeks after N application sampling time (130 mg p-nitrophenyl released kg^−1^ soil h^−1^).

#### 3.2.2. Lamberton

β-Glucosidase activity at Lamberton was not significantly affected by treatment but was affected by sampling time, year, and the year × sampling time interaction (Table 2). In 2014, β-glucosidase activity was greatest at the Pre-V6 sampling time (930 mg of p-nitrophenyl released kg^−1^ soil h^−1^) and least for samples collected at Post-V6 (421 mg of p-nitrophenyl released kg^−1^ soil h^−1^) (Table 5). After the Post-V6 sampling, β-glucosidase activity increased and reached a maximum 687 mg of p-nitrophenyl released kg^−1^ soil h^−1^ by PM. In 2015, there were fewer differences in β-glucosidase activity among soil sampling times, and soil samples collected at 3wk-R1 had greater β-glucosidase activity (895 mg of p-nitrophenyl released kg^−1^ soil h^−1^) than all other sampling times. In 2016, β-glucosidase activity in soil samples taken at Post-V6 and Post-R1 (mean = 691 mg of p-nitrophenyl released kg^−1^ soil h^−1^) were greater than those of samples collected at Pre-V6 and Pre-R1 (mean = 605 mg of p-nitrophenyl released kg^−1^ soil h^−1^) and at 3wk-V6 and 3wk-R1 (mean = 611 mg of p-nitrophenyl released kg^−1^ soil h^−1^).

#### 3.2.3. Waseca

There was no significant effect of treatment on β-glucosidase activity at Waseca; however, the effects of sampling time and the year × sampling time interaction were significant (Table 2). In 2014, β-glucosidase activity at the beginning of the season (Pre-V6) was 993 mg of p-nitrophenyl kg^−1^ soil h^−1^, greater than that at all other sampling times (Table 5). β-Glucosidase activity decreased sharply by the next sampling at Post-V6 to 611 mg of p-nitrophenyl released kg^−1^ soil h^−1^, followed by a slow increase through PM (817 mg of p-nitrophenyl released kg^−1^ soil h^−1^). In 2015, there were no changes in β-glucosidase activity throughout the growing season, with the exception that it was lower at Post-R1 than at PM, with 822 and 929 mg of p-nitrophenyl released kg^−1^ soil h^−1^, respectively (Table 5). In 2016, β-glucosidase activity fluctuated throughout the season, with the lowest concentration at Pre-V6, 722 mg of p-nitrophenyl released kg^−1^ soil h^−1^. At Post-V6, β-glucosidase activity had reached the second-highest level of the season (830 mg of p-nitrophenyl released kg^−1^ soil h^−1^). β-Glucosidase activity after Post-V6 decreased and remained low until 3wk-R1, when it increased to levels similar to those at Post-V6 (Table 5).

### 3.3. Acid Phosphatase Activity

#### 3.3.1. Becker

No significant differences due to treatment, sampling time, year, or their interactions were observed for acid phosphatase activity at Becker (Table 2).

#### 3.3.2. Lamberton

Acid phosphatase activity at Lamberton was not significantly influenced by treatment effects but was affected by sampling time, year, and the year × sampling time interaction (Table 2). In 2014, acid phosphatase activity was greater in soil samples collected prior to the in-season fertilizer applications at Pre-V6 and Pre-R1, averaging 2146 and 2105 mg of p-nitrophenyl released kg^−1^ soil h^−1^, respectively (Table 6). In 2014, acid phosphatase activity decreased after the V6 and R1 fertilizer applications. Acid phosphatase activity also decreased following the R1 N application in 2015. In 2016, acid phosphatase activity at 3wk-V6, 3wk-R1, and PM ranged from 1903 to 1926 mg of p-nitrophenyl released kg^−1^ soil h^−1^, being greater than any other sampling time and lowest at Post-R1, 1406 mg of p-nitrophenyl released kg^−1^ soil h^−1^. Phosphatase activity was greater in 2015 (2010 mg of p-nitrophenyl released h^−1^ soil kg^−1^) than in 2014 and 2016 (1748 and 1746 mg of p-nitrophenyl released h^−1^ soil kg^−1^, respectively).

#### 3.3.3. Waseca

There was no significant effect of treatment on phosphatase activity at Waseca, but there were significant effects for sampling time, year, and the year × sampling time interaction (Table 2). In 2014, soil samples collected at Pre-V6 (1644 mg of p-nitrophenyl released kg^−1^ soil h^−1^) showed greater acid phosphatase activity than soil samples taken Post-V6, 3wk-V6, 3wk-R1, and PM (Table 6). Phosphatase activity was greater in 2015 (1605 mg of p-nitrophenyl released h^−1^ soil kg^−1^) than in 2014 and 2016 (1320 and 1419 mg of p-nitrophenyl released h^−1^ soil kg^−1^, respectively).

### 3.4. Arylsulfatase Activity

#### 3.4.1. Becker

There were significant effects of treatment, sampling time, year, and the interaction between sampling time and year on arylsulfatase activity at Becker (Table 2). Arylsulfatase activity with the non-N-fertilized control (93 mg of p-nitrophenyl released kg^−1^ soil h^−1^) was greater than that with the F100, F125, PP100, PP125, and Sp100 treatments (mean = 76 mg of p-nitrophenyl released kg^−1^ soil h^−1^) (Table 4). Arylsulfatase activity tended to decrease as the N application increased, and the treatments that received the largest N rate (F125 and PP125) had the lowest values (mean = 71 mg of p-nitrophenyl released kg^−1^ soil h^−1^) (Table 4).

In 2014, arylsulfatase activity was greater at the Pre-V6 sampling time (138 mg of p-nitrophenyl released kg^−1^ soil h^−1^) than at any other sampling time (mean = 88 mg of p-nitrophenyl released kg^−1^ soil h^−1^), with no differences among the remaining sampling times (Table 7). In 2015, an opposite trend was observed, and arylsulfatase activity was greater from Post-R1 to PM (mean = 115 mg of p-nitrophenyl released kg^−1^ soil h^−1^) compared with the early samplings (mean = 76 mg of p-nitrophenyl released kg^−1^ soil h^−1^). In 2016, arylsulfatase activity had more variability throughout the season, being greater in the first and last sampling times compared with the mid-season samplings (Table 7).

#### 3.4.2. Lamberton

Treatment, sampling time, year, and the sampling time × year interaction significantly affected arylsulfatase activity at Lamberton (Table 2). Arylsulfatase activity was greater for treatments receiving no N or lower rates of N (Table 4).

In 2014, arylsulfatase activity was greater at Pre-V6 (224 mg of p-nitrophenyl released kg^−1^ soil h^−1^) than at any other sampling time (Table 7). For the remainder of the season, arylsulfatase activity was stable, with the exception of a significant high arylsulfatase activity observed at the 3wk-R1 sampling. In 2015, a similar behavior was observed compared with 2014; arylsulfatase activity of samples collected at Pre-V6 (179 mg of p-nitrophenyl released kg^−1^ soil h^−1^) was greater than that of samples collected at later times, with the exception of a significant high arylsulfatase activity observed at 3wk-R1 (210 mg of p-nitrophenyl released kg^−1^ soil h^−1^). In 2016, the results were similar to those observed in 2014 and 2015, where arylsulfatase activity was highest at the first sampling (160 mg of p-nitrophenyl released kg^−1^ soil h^−1^). In 2016, there was also an increase in arylsulfatase activity at the Post-R1 sampling time (162 mg of p-nitrophenyl released kg^−1^ soil h^−1^).

#### 3.4.3. Waseca

Arylsulfatase activity was not significantly influenced by treatment but was affected by sampling time and the year × sampling time interaction (Table 2). In 2014, arylsulfatase activity taken in soil samples at Pre-V6 and Post-V6 (mean = 645 mg of p-nitrophenyl released h^−1^ soil kg^−1^) was greater than at all other sampling times (mean = 423 mg of p-nitrophenyl released kg^−1^ soil h^−1^) (Table 7). In 2015, there were no differences in arylsulfatase activity among sampling times. In 2016, there was a large spike in arylsulfatase activity at 3wk-V6 (887 mg of p-nitrophenyl released kg^−1^ soil h^−1^), which was greater than all other sampling times (263 to 464 mg of p-nitrophenyl released kg^−1^ soil h^−1^).

### 3.5. Principal Component Analysis

Principal component analysis was carried out to determine the relationships between soil enzyme activity, soil nitrogen, and DM yield. Eigenvalues from the principal component analysis indicate that the first four principal components accounted for 66.08% of the variance of data (Factor 1: 22.9%, Factor 2: 21.0%, Factor 3: 12.05%, and Factor 4: 10.13%). β-Glucosidase activity (PNG) had a positive effect on Factor 1 (>0.900), while sampling time and maize dry matter yield had a positive effect on Factor 2 (>0.900) (Table 8).

Pearson’s correlation analysis suggests that there was no correlation between soil enzyme activity and soil NO_3_, soil NH_4_, and maize dry matter yield (Table 9). However, within soil enzymes, there was a positive correlation of 63.4% between β-glucosidase and arylsulfatase and a 70.7% positive correlation between β-glucosidase and acid phosphatase. The correlation between β-glucosidase and arylsulfatase and between β-glucosidase and acid phosphatase is likely driven by soil organic C [22,23]. The lack of a significant correlation between soil enzymes and soil nitrogen and maize dry matter yield is additional confirmation that the rate and timing of N fertilization had little impact on the activity of FDA, β-glucosidase, acid phosphatase, and arylsulfatase.

## 4. Discussion

### 4.1. FDA

Fluoroscein diacetate hydrolysis was not affected by the N rate and time of application at any location, but it did fluctuate across the growing season and was affected by year (Table 2, Table 4 and Table 5). For example, FDA activity across all treatments at Becker was greater one and three weeks after the respective N applications. Increased microbial activity after N application could be due to increased root growth and, thus, C inputs through root biomass and exudates favoring microbial growth [11,34,35]. Given the low soil OM content at Becker (13 g kg^−1^), increases in soil C input would likely improve soil microbial activity [35,36]. The decrease in FDA from 2014 to 2016 at all three sites could be a result of continuous maize production and/or continued N fertilization. Monocropping systems have been reported to negatively impact the metabolic activity of microbes in the soil [35]. Long-term field experiments have shown that microbial C and N were lower in fields with continuous maize than in fields with crop rotation, and the responses were not influenced by N application (Moore et al. 2000). A decline in microbial activity with continuous maize compared to crop rotation may be due to factors such as reduction in easily decomposable organic compounds returned to the soil, less root density leading to lower root exudates, a less stable microclimate, and poor soil structure (e.g., aggregate stability) [2,35,37]. In contrast, microbial biomass carbon has been reported to decrease after N fertilization by as much as 9.5% compared with non-fertilized controls [36] and also when more than 180 kg N ha^−1^ is used for over five years [38]. Nitrogen fertilization could indirectly decrease microbial biomass by increasing soil hydrolase activities, which might act as a proxy for microbial biomass [11]. The enzyme that represents soil hydrolase activity investigated in this study was β-glucosidase. Figure 1 shows the negative relationship between FDA and β-glucosidase activity observed in this study. The data presented in Figure 1 are the average over all treatments and sampling times for each location in each year. Therefore, there are three points per location. It is possible that a combination of reduced plant diversity and, with it, reduced diversity in root exudates, in addition to increased soil hydrolase activity, could have caused a shift in the microbial community in the soil leading to a lower FDA activity as observed in this study.

### 4.2. Β-Glucosidase Activity

In our study, β-glucosidase activity in the first year after soybean was always greater at the Pre-V6 sampling than at the Post-V6 sampling (Table 4). This result suggests that the diversity of residue early in the season had a greater impact on β-glucosidase activity than N application. This agrees with literature data, which show that diversity in residue increases β-glucosidase activity [37,39,40]. In addition, planting different crops during a rotation can increase β-glucosidase activity, as it has been reported to be greater with a four-year rotation than in continuous maize [41,42]. It is likely that the diverse cropping systems provide diverse residue in addition to diverse root growth and root exudates, leading to a more dynamic and diversified microbial community [35]. The lack of an effect of N fertilizer rate or timing of application on β-glucosidase activity, with a significant response to residue only in the first sampling at the first year at all locations, suggests that β-glucosidase is more susceptible to changes in carbon than to changes in available N in the soil. Furthermore, β-glucosidase has been regarded as an enzyme that is very sensitive to changes in management [6], for example, switching from a soybean-maize rotation to continuous maize or moving from a no-till system to some form of tillage. Other research has also reported that β-glucosidase activity does not change as a result of N application before maize [41]. However, in some instances, N fertilizer has been found to increase β-glucosidase activity [4,11,43]. Although β-glucosidase was affected by year in this study, this effect was mainly due to an increase in β-glucosidase activity in 2015 (612 mg of p-nitrophenyl released kg^−1^ soil h^−1^) compared with 2014 and 2016 (523 and 537 mg of p-nitrophenyl released kg^−1^ soil h^−1^, respectively). Other studies have failed to find differences in β-glucosidase activity in continuous maize over the years [44].

### 4.3. Acid Phosphatase Activity

Acid phosphatase was the enzyme least affected by N fertilizer rate, time of application, and year. At Waseca, differences among sampling times only occurred in 2014. At Lamberton, N application often decreased acid phosphatase activity soon after application; however, in the early sampling times it tended to increase after this initial drop and then remain low during the rest of the season. In contrast, N application has been reported to be directly linked to soil phosphatase activity in soils from different regions of the planet [45,46]. Differences among sampling dates could have been due to soil moisture and temperature [47], along with soil pH and phosphorus availability [48,49]. The dominant influence of sampling date on soil properties, including alkaline phosphomonoesterase activity, was also reported by Shi et al. [49] from an 18-year maize-soybean rotation in eastern Canada. The lack of differences in acid phosphatase activity between sampling times in 2015 and 2016 at Waseca corresponds with findings from other studies where the application of N fertilizer was not shown to affect soil acid phosphatase activity [16,17,47]. Phosphatase activity has been shown to be correlated with organic C, and organic C could have an important role in protecting and maintaining acid phosphatase activity [22].

### 4.4. Arylsulfatase Activity

Arylsulfatase activity was affected by N application at Becker and Lamberton. At Becker, N application reduced arylsulfatase activity in comparison with the control plots, which showed the greatest levels of arylsulfatase activity. The results at Lamberton were not as clear-cut; the control plot did tend to have the greatest arylsulfatase activity compared with other treatments, but the differences were only significant when compared with the PP125, Sp75, and TSp100 treatments. While some studies have also reported that N addition decreased arylsulfatase activity by 39.6% [21,23], others have shown that arylsulfatase remains unchanged after N fertilization [45].

At Lamberton, soil moisture seemed to have had a more pronounced effect on arylsulfatase activity than N fertilization. A spike in arylsulfatase activity was often preceded by precipitation in the days leading up to the sampling date. At Lamberton in 2014, there was a sharp increase in arylsulfatase activity at the 3wks-R1 sampling; this increase in activity could have been a result of the 14 mm of precipitation that occurred two to five days before sampling. In 2015, a similar spike was observed at the 3wks-R1 sampling, where 8.1 mm of rain fell three days before the sampling date. The addition of water to soils has also resulted in increased arylsulfatase activity [21]. Water addition may have led to increased arylsulfatase activity by increasing soluble C in the soil solution, as increased precipitation has been found to increase labile SOC at 0–10 cm depth significantly [50]. Macro- and micro-climatic changes have been shown to influence microbial activity, particularly when soil pore space is one-half to two-thirds saturated [51]. However, microbial activity will decrease in anaerobic or near anaerobic conditions [51]. Arylsulfatase activity has been found to be greater in the spring after snowmelt or in the rainy season than during the dry season, when precipitation is decreased [51,52]. Furthermore, arylsulfatase activity has been reported to be greater in cropping systems that provide a greater amount of soluble C to the microbial population present at any given moment [23].

In general, the results of this research showed that the addition of N does not have a significant and consistent impact on soil enzyme activity, although sometimes the addition of N did cause a significant change. We found that having residue from different crops or soil moisture (rainfall) played a much larger role in soil microbial activity (as measured by enzyme activity) than N application. This result seems to suggest that continuous corn has the potential, in the long run, to limit soil function by reducing microbial activity.

## 5. Conclusions

Nitrogen application to continuous maize production was found to have small effects on soil enzyme activities. Split application of N was found to increase FDA activity at the Waseca site, and the application of N was found to reduce arylsulfatase activity at the Becker site. Collecting soil samples for enzyme activity tests during the growing season showed that enzyme activity varied, in most cases randomly, throughout the year and on a year-to-year basis. For example, samples collected after N application at Becker were found to have greater FDA activity than samples collected prior to N application. In contrast, at Lamberton and Waseca, sampling after N application was found to show lower FDA activity than samples collected prior to N application in some years, but the results were not consistent across all years and treatments. A common trend observed at all locations was a decline in FDA and β-glucosidase activity from 2014 to 2016, suggesting that the soybean residue provided ideal conditions for high FDA and β-glucosidase activity at all locations. The decline in FDA and β-glucosidase activity from 2014 to 2016 suggests that crop rotation could be beneficial at Becker, Lamberton, and Waseca for increased soil enzyme activity. Phosphatase activity was the least affected by the parameters measured in this study. The results of this study indicate that arylsulfatase activity was affected by soil moisture content and rainfall more than by N application, sampling time, and year. Overall, the variability of soil enzyme activity under maize in field conditions suggests that urea N application has a small effect on enzyme activity. The lack of enzyme response to N application in combination with higher enzyme activity when corn followed soybean in the first year of the study suggests that some form of organic N could have a larger impact on microbial activity.

## Figures and Tables

**Figure 1 plants-11-02247-f001:**
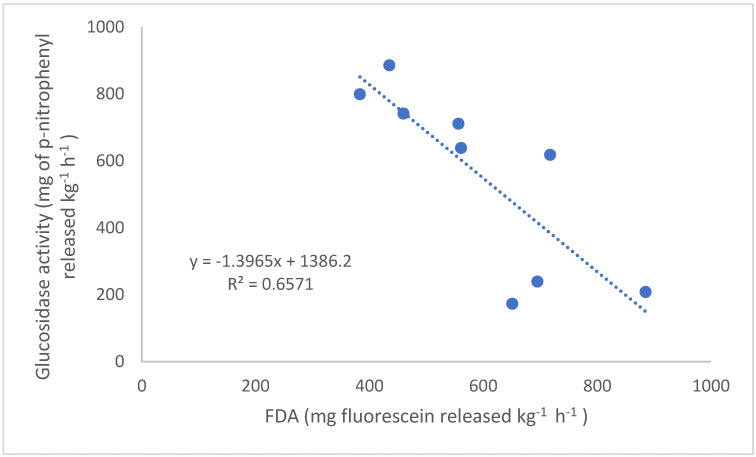
Relationship between the yearly average of fluorescein diacetate hydrolysis and ß_-1_,4-glucosidase from 2014 to 2016 across Becker, Waseca, and Lamberton.

**Table 1 plants-11-02247-t001:** Treatments tested in the study. Nitrogen rates were based on the University of Minnesota guidelines (Kaiser et al., 2011).

Treatment	N Applied (% of Recommended)	Time of Application
Control	0	N/A
F100	100	Fall
F125	125	Fall
S100	100	Pre-plant Spring
S125	125	Pre-plant Spring
Sp75	75	One half of total amount Pre-plant in the Spring and the second half at V6
Sp100	100	One half of total amount Pre-plant in the Spring and the second half at V6
TSp50	50	One third of total amount Pre-plant in the Spring, one third at V6, and the last third at R1
TSp75	75	One third of total amount Pre-plant in the Spring, one third at V6, and the last third at R1
TSp100	100	One third of total amount Pre-plant in the Spring, one third at V6, and the last third at R1

**Table 2 plants-11-02247-t002:** Tests of fixed effects of N fertilizer rate and timing of application on FDA hydrolysis, β-glucosidase activity, phosphatase activity, and arylsulfatase activity from at Becker, Lamberton, and Waseca, MN.

Effect	FDA Hydrolysis			Β-glucosidase Activity			Phosphatase Activity			Arylsulfatase Activity		
	Becker	Lamberton	Waseca	Becker	Lamberton	Waseca	Becker	Lamberton	Waseca	Becker	Lamberton	Waseca
	*Pr > F*											
Treatment (T)	NS ^†^	NS	*	NS	NS	NS	NS	NS	NS	**	NS	NS
Sampling time (S)	***	***	***	***	***	***	NS	***	***	***	***	***
Year (Y)	*	*	***	*	***	*	NS	***	***	*	*	NS
T × S	NS	NS	NS	NS	NS	NS	NS	NS	NS	NS	NS	NS
T × Y	NS	NS	NS	NS	NS	NS	NS	NS	NS	NS	NS	NS
S × Y	NS	***	***	***	***	***	NS	***	***	***	***	***
T × S × Y	NS	NS	NS	NS	NS	NS	NS	NS	NS	NS	NS	NS

* Significant at the 0.05 probability level. ** Significant at the 0.01 probability level. *** Significant at the 0.001 probability level. ^†^ NS, nonsignificant.

**Table 3 plants-11-02247-t003:** In-season fluorescein diacetate (FDA) hydrolysis at Becker, Lamberton, and Waseca.

Sampling Time ^a^	Becker	Lamberton			Waseca		
	Averaged over All Years	2014	2015	2016	2014	2015	2016
	——————mg of fluorescein released kg^−1^ h^−1^ ——————						
Pre-V6	703 c *	931 a *	656 cde	540 efg	543 b *	618 a	392 fgh
Post-V6	769 b	636 cdef	557 efg	524 fg	434 def	443 cde	410 ef
3 wks-V6	931 a	521 fg	534 efg	605 cdefg	401 efg	433 def	438 cdef
Pre-R1	703 c	865 b	593 defg	615 cdef	441 cde	391 fgh	-
Post-R1	780 b	720 c	534 efg	603 cdefg	479 cd	482 c	335 ij
3 wks-R1	668 d	692 cd	534 efg	520 fg	459 cd	355 hij	360 ghi
PM	641 e	657 cde	487 g	520 fg	461 cd	322 j	364 ghi

**^a^** Pre-V6, soil samples taken 1 week before V6 urea applications; Post-V6, soil samples taken 1 week after V6 urea applications; 3wk-V6, soil samples taken 3 weeks after V6 urea applications; Pre-R1, soil samples taken 1 week before R1 urea application; Post-R1, soil samples taken 1 week after R1 urea applications; 3 wks-R1, soil samples taken 3 weeks after R1 urea applications; PM, soil samples taken at physiological maturity. * Means followed by different lowercase letters are significantly different at the 0.05 probability level.

**Table 4 plants-11-02247-t004:** Treatment effect on fluorescein diacetate (FDA) hydrolysis at Waseca and on arylsulfatase activity at Becker and Lamberton.

Treatment	Waseca	Becker	Lamberton
	mg of fluorescein released kg^−1^ h^−1^	mg of p-nitrophenyl released kg^−1^ h^−1^	
C ^a^	433 abc *	93 a *	175 a **
F100	420 bc	81 bcd	167 abc
F125	425 bc	72 d	156 abcd
PP100	402 c	76 cd	163 abcd
PP125	410 c	71 d	147 cd
Sp75	456 ab	89 ab	147 d
Sp100	467 a	80 bcd	166 abcd
TSp50	421 bc	92 ab	171 ab
TSp75	419 c	85 abc	157 abcd
TSp100	422 bc	89 ab	153 bcd

**^a^** C, control no N applied; F100, N applied in the fall at 100% of the recommended rate; F125, N applied in the fall at 125% of the recommended rate; PP100, N applied at pre-plant in the spring at 100% of the recommended rate; PP125, N applied at pre-plant in the spring at 125% of the recommended rate; Sp75, half of the N applied at pre-plant in the spring and half at V6 at 75% of the recommended rate; Sp100, half of the N applied at pre-plant in the spring and half at V6 at 100% of the recommended rate; TSp50, a third of the N applied at pre-plant in the spring, a third at V6, and a third at R1 at 50% of the recommended rate; TSp75, a third of the N applied at pre-plant in the spring, a third at V6, and a third at R1 at 75% of the recommended rate; TSp100, a third of the N applied at pre-plant in the spring, a third at V6, and a third at R1 at 100% of the recommended rate. * Means followed by different lowercase letters are significantly different at the 0.05 probability level. ** Means followed by different lowercase letters are significantly different at the 0.10 probability level.

**Table 5 plants-11-02247-t005:** In-season β-glucosidase activity across treatments at Becker, Lamberton, and Waseca from 2014 to 2016.

Scheme ^a^	Becker			Lamberton			Waseca		
	2014	2015	2016	2014	2015	2016	2014	2015	2016
	mg of p-nitrophenyl Released kg^−1^ h^−1^								
Pre-V6	281 ab *	246 bc	190 efgh	930 a *	682 bcde	581 g	993 a *	910 abc	722 gh
Post-V6	158 hi	306 a	195 defgh	421 i	693 bc	694 bc	611 i	894 abcd	830 bcde
3 wks-V6	180 fgh	217 def	123 i	494 h	666 bcdef	589 g	667 h	845 bcde	822 cdef
Pre-R1	198 defg	198 defg	172 gh	520 h	706 b	628 ef	673 h	896 abcd	
Post-R1	217 cdef	225 cde	172 gh	649 cdef	640 def	688 bcd	672 h	822 cdef	733 fgh
3 wks-R1	223 cde	249 bc	138 i	625 fg	895 a	632 ef	756 efg	902 abcd	804 defg
PM	201 defg	234 cd	221 cdef	687 bcd	695 bc	657 bcdef	817 cdefg	929 ab	883 bcd

**^a^** Pre-V6, soil samples taken 1 week before V6 urea applications; Post-V6, soil samples taken 1 week after V6 urea applications; 3wk-V6, soil samples taken 3 weeks after V6 urea applications; Pre-R1, soil samples taken 1 week before R1 urea application; Post-R1, soil samples taken 1 week after R1 urea applications; 3 wks-R1, soil samples taken 3 weeks after R1 urea applications; PM, soil samples taken at physiological maturity. * Means followed by different lowercase letters are significantly different at the 0.05 probability level.

**Table 6 plants-11-02247-t006:** In-season acid phosphatase activity across treatments at Lamberton and Waseca from 2014 to 2016.

Sampling Time ^a^	Lamberton			Waseca		
	2014	2015	2016	2014	2015	2016
	mg of p-nitrophenyl released kg^−1^ h^−1^					
Pre-V6	2146 a *	2159 a	1736 c	1644 ab *	1709 ab	1504 abc
Post-V6	1448 e	2175 a	1713 cd	1232 c	1739 ab	1468 abc
3 wks-V6	1721 cd	1879 b	1926 b	1123 c	1533 abc	1499 abc
Pre-R1	2105 a	2166 a	1631 d	1573 abc	1597 abc	-
Post-R1	1892 b	1903 b	1406 e	1338 abc	1484 abc	1293 bc
3 wks-R1	1508 e	1917 b	1903 b	1171 c	1771 a	1437 abc
PM	1417 e	1869 b	1907 b	1158 c	1400 abc	1314 abc

**^a^** Pre-V6, soil samples taken 1 week before V6 urea applications; Post-V6, soil samples taken 1 week after V6 urea applications; 3wk-V6, soil samples taken 3 weeks after V6 urea applications; Pre-R1, soil samples taken 1 week before R1 urea application; Post-R1, soil samples taken 1 week after R1 urea applications; 3 wks-R1, soil samples taken 3 weeks after R1 urea applications; PM, soil samples taken at physiological maturity. * Means followed by different lowercase letters are significantly different at the 0.05 probability level.

**Table 7 plants-11-02247-t007:** In-season arylsulfatase activity at Becker, Lamberton, and Waseca from 2014 to 2016.

Sampling Time ^a^	Becker			Lamberton			Waseca		
	2014	2015	2016	2014	2015	2016	2014	2015	2016
	mg of p-nitrophenyl released kg^−1^ soil h^−1^								
Pre-V6	138 a *	70 cdefgh	70 cdefg	224 a *	179 cd	160 defg	678 ab*	581 abcd	346 cde
Post-V6	91 bc	83 cdefg	59 h	173 de	155 efgh	136 ghi	611 abc	528 bcde	296 de
3 wks-V6	86 cdef	67 efgh	61 gh	166 de	132 i	127 i	472 bcde	478 bcde	887 a
Pre-R1	86 cdef	68 defgh	64 fgh	168 de	161 defg	133 hi	446 bcde	429 bcde	-
Post-R1	86 cdef	91 bc	58 h	160 defg	119 i	162 def	365 cde	335 cde	263 e
3 wks-R1	90 bcd	143 a	55 h	195 bc	210 ab	140 fghi	375 cde	546 bcde	265 e
PM	89 cde	112 b	74 cdefg	163 def	177 cd	124 i	455 bcde	574 abcd	464 bcde

**^a^** Pre-V6, soil samples taken 1 week before V6 urea applications; Post-V6, soil samples taken 1 week after V6 urea applications; 3wk-V6, soil samples taken 3 weeks after V6 urea applications; Pre-R1, soil samples taken 1 week before R1 urea application; Post-R1, soil samples taken 1 week after R1 urea applications; 3 wks-R1, soil samples taken 3 weeks after R1 urea applications; PM, soil samples taken at physiological maturity. * Means followed by different lowercase letters are significantly different at the 0.05 probability level.

**Table 8 plants-11-02247-t008:** Component matrix by factor analysis for soil enzyme activity, soil nitrogen, and maize dry matter yield.

	Factor 1	Factor 2	Factor 3	Factor 4
Year	0.061	0.134	−0.738	0.555
Sampling time	−0.001	0.946	0.135	−0.019
Block	0.002	−0.039	0.513	0.762
Fluorescein diacetate	−0.599	−0.278	0.482	0.026
Arylsulfatase	0.722	−0.082	0.165	−0.046
Acid phosphatase	0.704	−0.115	0.140	−0.054
β-glucosidase	0.941	−0.009	0.140	0.010
Soil NO_3_	0.149	−0.466	−0.179	0.204
Soil NH_4_	−0.032	0.004	−0.204	−0.267
Maize dry matter yield	0.048	0.934	0.078	0.072

**Table 9 plants-11-02247-t009:** Pearson’s correlation coefficients among soil enzyme activity, soil nitrogen, and maize dry matter yield (*n* = 2473).

	Year	Time	Block	Fluorescein Diacetate	Arylsulfatase	Acid Phosphatase	β-Glucosidase	Soil NO_3_	Soil NH_4_	Maize Dry Matter Yield
Year	1.000	0.002	0.000	−0.261	−0.102	−0.025	0.004	0.042	0.026	0.087
Time		1.000	−0.001	−0.147	−0.060	−0.080	0.014	−0.301	0.027	0.905
Block			1.000	0.165	0.086	−0.013	0.050	0.002	−0.027	0.012
Fluorescein diacetate				1.000	−0.361	−0.174	−0.395	0.034	0.000	−0.175
Arylsulfatase					1.000	0.172	0.634	0.047	0.009	−0.075
Acid phosphatase						1.000	0.707	0.109	−0.027	−0.027
β-glucosidase							1.000	0.117	−0.035	0.060
Soil NO_3_								1.000	0.021	−0.221
Soil NH_4_									1.000	−0.020
Maize dry matter yield										1.000

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
