# Peer review of "Soil Enzyme Activity Behavior after Urea Nitrogen Application"

_plants, 2022, doi:10.3390/plants11172247_

Round 1
Reviewer 1 Report
This manuscript describes the time-profiling of soil enzyme activities after relatively long-term (2014-2016 year) N fertilization application to soil. The authors employed urea as N fertilizer and also investigated four enzyme activities including FDA hydrolysis, beta-glucosidase, acid-phosphomonoesterase, and arylsulfatase. Finally, this manuscript suggested that urea N application could not affect the enzyme activities, and other organic N source could be more crucial source for the corresponding enzyme activities instead.
Overall, the manuscript seems to be well written with logical experimental design and proper methods, followed by the results and discussion to be properly presented. However, the current manuscript requires some revisions to improve the quality and clarify some weak pointes before the acceptance in Plants journal.
#1. As I understood the current study, the climate information would be key factor for the enzyme activities, because the growing season make the enzyme activities to be varied. Thus, it recommends that the relationship between the climate information and enzyme activities could be needed to be discussed.
#2. Line 25-28: In abstract, the current summary just includes the requirement of other organic N source instead of urea N. It could be recommended that the authors added more contents including examples of other N sources with mentioning why other N sources should be needed, for the readers to be easily understand.
#3. Line 74-148: The corresponding contents in Introduction part includes the descriptions of three soil enzymes, which seems to be so lengthy. Thus, please shorten the contents and move some parts to discussion part.
#4. Line 159-162: This content represents the information of FDA (hydrolysis), which seems to be written with being relatively short compared to other three enzymes. Thus, please fortify the introduction for FDA (hydrolysis).
#5. The authors mentioned the site description with sampling information with just texting in materials and methods part. However, it could be better to add the figure showing the map for site description and the real photo for the sampling information.
#6. When the g force is mentioned, the multiplication sign (X) should be written, not English letter. Check throughout the whole text in manuscript.
#7. When the wavelength is mentioned, the space should be inserted between the number and unit (nm). Check throughout the whole text in manuscript.
#8. Line 288: For the contents of Becker, the authors should insert the sub-section number (3.1.1). In this regard, you have to change the whole sub-section number after 3.1.1.
#9. The current table includes the lower-case codes for statistical analysis. However, these lower-case letters should be indicated in superscript to be easily represented. Check throughout all the tables.
#10. Line 684-688: This conclusion seems to be mismatched with the summary in abstract part. Please rewrite the final conclusion in this section with adding the similar content as summary in abstract section.
#11. Please revise the information of author affiliations and email addresses, and also add the author contribution, IRB statement, acknowledgments, conflicts of interest, etc, after conclusion part. Please check the recent articles in plants journal.
Reviewer 2 Report
This study examined the effect of different nitrogen rates and the timing of application in maize on soil enzyme activities. Results contribute to the growing concern for using urea in agricultural crops and how it affects soil enzymatic activity. This study is timely and demand-driven.
The abstract was presented very well. The introduction is comprehensive and well-referenced. The methods used are robust, and presented in a detailed way possible. Statistical analyses used are fitting for the study. The results were presented in detail but need improvement. The authors have exhaustively dug into the mechanisms and drivers in the Discussions Section and use previous studies to support their claims. The manuscript ends with well-described highlights, although it also needs slight improvement.
Hereunder are my specific comments:
Lines 149-162:
I suggest establishing here the novelty of the work. State the objectives and the hypotheses along with it, if there are any.
Methods lines 188-203:
I think this will be best presented in a Table format. The text looks repetitive of the many treatments used, percentages, and rates, so it is to better to show them in the table.
Results, Table 2:
If Becker's site has been averaged over the years, then why would other sites not be averaged for consistency and uniformity in the analysis? Otherwise, the average per year in the Becker site must also be spread similar to other sites. However, I think it would be best to do the averaging per site (covering all years) to get the overall mean for each location.
Line 511: 3.5 The Principal Component Analysis
I think showing the PCA circle to easily see the correlations among variables will be more informative. In the methods, the authors mentioned the Akaike weights. Were they able to determine which treatments or variables (a single or combination) can best explain the variations in soil enzyme activity, soil nitrogen, and DM yield using the result from the AIC?
Figure 1:
This is an excellent result to support your claim about the relationship between FDA and specific enzyme activity and should be part of the Results section. Please make the units in Figure 1 correctly written.
Discussion Section (General):
I suggest the authors add a section to briefly provide the implication of their results and how they will contribute to corn farming or the science community.
Conclusion, Lines 76-78:
In the Conclusion Section, please avoid presenting statements that were not covered in your study or not a result of your experiment. Only provide the most highlighted results and their implications. Reiterate the novelty of your work here and boost your work. Provide the readers a take-home message about your work.
General for Tables:
Footnotes in tables are lengthy, maybe incorporate them in the method section, so you don’t have to describe them in tables again.
Round 2
Reviewer 1 Report
Thank you for your response and proper revision of your manuscript according to my concerns. Thus, I recommend this revised manuscript deserves the acceptance in Plants journal.